# Neural Parameter Allocation Search

**Bryan A. Plummer\*[†], Nikoli Dryden\*[‡], Julius Frost[†], Torsten Hoefler[‡], Kate Saenko**[†§]
[†]Boston University, [‡]ETH Zürich, [§]MIT-IBM Watson AI Lab
[†]{bplum,juliusf,saenko}@bu.edu
[‡]{nikoli.dryden,torsten.hoefler}@inf.ethz.ch

## Abstract

Training neural networks requires increasing amounts of memory. Parameter sharing can reduce memory and communication costs, but existing methods assume networks have many identical layers and utilize hand-crafted sharing strategies that fail to generalize. We introduce Neural Parameter Allocation Search (NPAS), a novel task where the goal is to train a neural network given an arbitrary, fixed parameter budget. NPAS covers both low-budget regimes, which produce compact networks, as well as a novel high-budget regime, where additional capacity can be added to boost performance without increasing inference FLOPs. To address NPAS, we introduce Shapeshifter Networks (SSNs), which automatically learn where and how to share parameters in a network to support any parameter budget without requiring any changes to the architecture or loss function. NPAS and SSNs provide a complete framework for addressing generalized parameter sharing, and can also be combined with prior work for additional performance gains. We demonstrate the effectiveness of our approach using nine network architectures across four diverse tasks, including ImageNet classification and transformers.

## 1 Introduction

Training neural networks requires ever more computational resources, with GPU memory being a significant limitation (Rajbhandari et al., 2021). Method such as checkpointing (*e.g.*, Chen et al., 2016; Gomez et al., 2017; Jain et al., 2020) and out-of-core algorithms (*e.g.*, Ren et al., 2021) have been developed to reduce memory from activations and improve training efficiency. Yet even with such techniques, Rajbhandari et al. (2021) find that model parameters require significantly greater memory bandwidth than activations during training, indicating parameters are a key limit on future growth. One solution is cross-layer parameter sharing, which reduces the memory needed to store parameters, which can also reduce the cost of communicating model updates in distributed training (Lan et al., 2020; Jaegle et al., 2021) and federated learning (Konečný et al., 2016; McMahan et al., 2017), as the model is smaller, and can help avoid overfitting (Jaegle et al., 2021). However, prior work in parameter sharing (*e.g.*, Dehghani et al., 2019; Savarese & Maire, 2019; Lan et al., 2020; Jaegle et al., 2021) has two significant limitations. First, they rely on suboptimal hand-crafted techniques for deciding where and how sharing occurs. Second, they rely on models having many identical layers. This limits the network architectures they apply to (*e.g.*, DenseNets (Huang et al., 2017) have few such layers) and their parameter savings is only proportional to the number of identical layers.

To move beyond these limits, we introduce **Neural Parameter Allocation Search (NPAS)**, a novel task which generalizes existing parameter sharing approaches. In NPAS, the goal is to identify where and how to distribute parameters in a neural network to produce a high-performing model using an arbitrary, fixed parameter budget and no architectural assumptions. Searching for good sharing strategies is challenging in many neural networks due to different layers requiring different numbers of parameters or weight dimensionalities, multiple layer types (*e.g.*, convolutional, fully-connected, recurrent), and/or multiple modalities (*e.g.*, text and images). Hand-crafted sharing approaches, as in prior work, can be seen as one implementation of NPAS, but they can be complicated to create for complex networks and have no guarantee that the sharing strategy is good. Trying all possible permutations of sharing across layers is computationally infeasible even for small networks. To our knowledge, we are the first to consider automatically searching for good parameter sharing strategies.

---

*indicates equal contribution

| | Neural Parameter Allocation Search (ours) | Prior work in Cross-Layer Parameter Sharing | Knowledge Distillation | Pruning/Quantizing Networks |
|---|---|---|---|---|
| **makes training more efficient** | ☑ | ☑ | ☒ | ☒ |
| **supports any parameter budget (low *and* high)** | ☑ | ☒ | ☒ | ☒ |
| **generalizes to any architecture** | ☑ | ☒ | ☑ | ☑ |
| **focuses on improving task performance** | ☑ | ☒ | ☑ | ☒ |

Figure 1: **Comparison of related tasks.** Neural Parameter Allocation Search (NPAS) is a novel task where the goal is to train a neural network given a fixed parameter budget. This generalizes prior work, which supports only a subset of NPAS's settings. *E.g.*, pruning can decrease network parameters, but starts from a full network, and many cross-layer parameter sharing works rely on hand-crafted strategies that work for only a limited set of architectures. See Section 2 for discussion.

By supporting arbitrary parameter budgets, NPAS explores two novel regimes. First, while prior work considered using sharing to reduce the number of parameters (which we refer to as low-budget NPAS, LB-NPAS), we can also *increase* the number of parameters beyond what an architecture typically uses (high-budget NPAS, HB-NPAS). HB-NPAS can be thought of as adding capacity to the network in order to improve its performance without changing its architecture (*e.g.*, without increasing the number of channels that would also increase computational time). Second, we consider cases where there are fewer parameters available to a layer than needed to implement the layer's operations. For such low-budget cases, we investigate parameter upsampling methods to generate the layer's weights.

A vast array of other techniques, including pruning (Hoefler et al., 2021), quantization (Gholami et al., 2021), knowledge distillation (Gou et al., 2021), and low-rank approximations (*e.g.*, Wu, 2019; Phan et al., 2020) are used to reduce memory and/or FLOP requirements for a model. However, such methods typically only apply at test/inference time, and actually are more expensive to train due to requiring a fully-trained large network, in contrast to NPAS. Nevertheless, these are also orthogonal to NPAS and can be applied jointly. Indeed, we show that NPAS can be combined with pruning or distillation to produce improved networks. Figure 1 compares NPAS to closely related tasks.

To implement NPAS, we propose *Shapeshifter Networks* (SSNs), which can morph a given parameter budget to fit any architecture by learning where and how to share parameters. SSNs begin by learning which layers can effectively share parameters using a short pretraining step, where all layers are generated from a single shared set of parameters. Layers that use parameters in a similar way are then good candidates for sharing during the main training step. When training, SSNs generate weights for each layer by down- or upsampling the associated parameters as needed.

We demonstrate SSN's effectiveness in high- and low-budget NPAS on a variety of networks, including vision, text, and vision-language tasks. *E.g.*, a LB-NPAS SSN implements a WRN-50-2 (Zagoruyko & Komodakis, 2016) using 19M parameters (69M in the original) and achieves an Error@5 on ImageNet (Deng et al., 2009) 3% lower than a WRN with the same budget. Similarity, we achieve a 1% boost to SQuAD v2.0 (Rajpurkar et al., 2016) with 18M parameters (334M in the original) over ALBERT (Lan et al., 2020), prior work for parameter sharing in Transformers (Vaswani et al., 2017). For HB-NPAS, we achieve a 1–1.5% improvement in Error@1 on CIFAR (Krizhevsky, 2009) by adding capacity to a traditional network. In summary, our key contributions are:

- We introduce Neural Parameter Allocation Search (NPAS), a novel task in which the goal is to implement a given network architecture using *any* parameter budget.
- To solve NPAS, we propose Shapeshifter Networks (SSNs), which automate parameter sharing. To our knowledge, SSNs are the first approach to automatically learn where and how to share parameters and to share parameters between layers of different sizes or types.
- We benchmark SSNs for LB- and HB-NPAS and show they create high-performing networks when either using few parameters or adding network capacity.
- We also show that SSNs can be combined with knowledge distillation and parameter pruning to boost performance over such methods alone.

## 2 NEURAL PARAMETER ALLOCATION SEARCH (NPAS)

In NPAS, the goal is to implement a neural network given a fixed parameter budget. More formally:

> **Neural Parameter Allocation Search** (NPAS): Given a neural network architecture with layers $\ell_1, \ldots, \ell_L$, which each require weights $w_1, \ldots, w_L$, and a fixed parameter budget $\theta$, train a high-performing neural network using the given architecture and parameter budget.

Any general solution to NPAS (*i.e.*, that works for arbitrary $\theta$ or network) must solve two subtasks:

1. Parameter mapping: Assign to each layer $\ell_i$ a subset of the available parameters.
2. Weight generation: Generate $\ell_i$'s weights $w_i$ from its assigned parameters, which may be any size.

Prior work, such as Savarese & Maire (2019) and Ha et al. (2016), are examples of weight generation methods, but in limited cases, *e.g.*, Savarese & Maire (2019) does not support there being fewer parameters than weights. To our knowledge, no prior work has automated parameter mapping, instead relying on hand-crafted heuristics that do not generalize to many architectures. Note weight generation must be differentiable so gradients can be backpropagated to the underlying parameters.

NPAS naturally decomposes into two different regimes based on the parameter budget relative to what would be required by a traditional neural network (*i.e.*, $\sum_i^L |w_i|$ versus $|\theta|$):

- Low-budget (LB-NPAS), with fewer parameters than standard networks ($\sum_i^L |w_i| < |\theta|$). This regime has traditionally been the goal of cross-layer parameter sharing, and reduces memory at training and test time, and consequentially reduces communication for distributed training.
- High-budget (HB-NPAS), with more parameters than standard networks ($\sum_i^L |w_i| > |\theta|$). This is, to our knowledge, a novel regime, and can be thought of as adding capacity to a network without changing the underlying architecture by allowing a layer to access more parameters.

Note, in both cases, the FLOPs required of the network do not significantly increase. Thus, HB-NPAS can significantly reduce FLOP overhead compared to larger networks.

The closest work to ours are Shared WideResNets (SWRN) (Savarese & Maire, 2019), Hypernetworks (HN) (Ha et al., 2016), and Lookup-based Convolutional Networks (LCNN) (Bagherinezhad et al., 2017). Each method demonstrated improved low-budget performance, with LCNN and SWRN focused on improving sharing across layers and HN learning to directly generate parameters. However, all require adaptation for new networks and make architectural assumptions. *E.g.*, LCNN was designed specifically for convolutional networks, while HN and SWRN's benefits are proportional to the number of identical layers (see Figure 3). Thus, each method supports limited architectures and parameter budgets, making them unsuited for NPAS. LCNN and HN also both come with significant computational overhead. *E.g.*, the CNN used by Ha et al. requires 26.7M FLOPs for a forward pass on a $32 \times 32$ image, but weight generation with HN requires an additional 108.5M FLOPs (135.2M total). In contrast, our SSNs require 0.8M extra FLOPs (27.5M total, $5\times$ fewer than HN). Across networks we consider, SSN overhead for a single image is typically 0.5–2% of total FLOPs. Note both methods generate weights once per forward pass, amortizing overhead across a batch (*e.g.*, SSN overhead is reduced to 0.008–0.03% for batch size 64). HB-NPAS is also reminiscent of mixture-of-experts (*e.g.*, Shazeer et al., 2017); both increase capacity without significantly increasing FLOPs, but NPAS allows this overparameterization to be learned without architectural changes required by prior work.

NPAS can be thought of as searching for efficient and effective underlying representations for a neural network. Methods have been developed for other tasks that focus on directly searching for more effective architectures (as opposed to their underlying representations). These include neural architecture search (*e.g.*, Bashivan et al., 2019; Dong & Yang, 2019; Tan et al., 2019; Xiong et al., 2019; Zoph & Le, 2017) and modular/self-assembling networks (*e.g.*, Alet et al., 2019; Ferran Alet, 2018; Devin et al., 2017). While these tasks create computationally efficient architectures, they do not reduce the number of parameters in a network during training like NPAS (*i.e.*, they cannot be used to train very large networks or for federated or distributed learning applications), and indeed are computationally expensive. NPAS methods can also provide additional flexibility to architecture search by enabling them to train larger and/or deeper architectures while keeping within a fixed parameter budget. In addition, the performance of any architectures these methods create could be improved by leveraging the added capacity from excess parameters when addressing HB-NPAS.

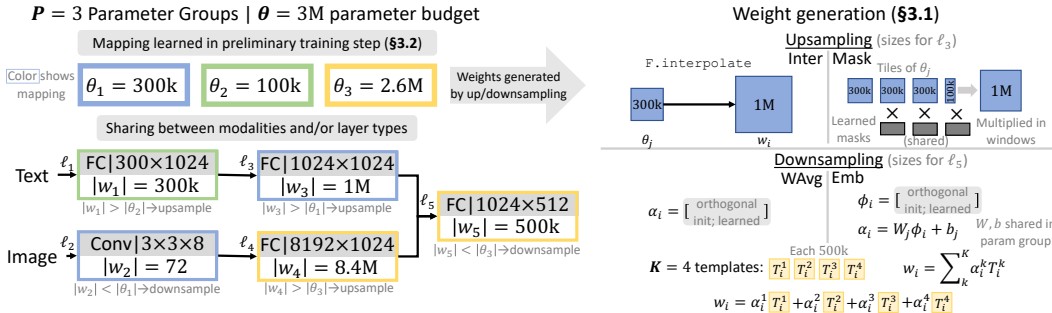

Figure 2: **Overview of Shapeshifter Networks (SSNs).** To train a SSN we begin by learning a mapping of layers to parameter groups (Section 3.2). Only layers within the same group share parameters. Then, during the forward pass each layer uses its shared parameters to generate the weights it needs for its operation (Section 3.1). Note that SSNs do not change a network's architecture or loss function, and automates the creation of a parameter sharing strategy for a network.

## 3 SHAPESHIFTER NETWORKS FOR NPAS

We now present Shapeshifter Networks (SSNs), a framework for addressing NPAS using generalized parameter sharing to implement a neural network with an arbitrary, fixed parameter budget. Figure 2 provides an overview and example of SSNs, and we detail each aspect below. An SSN consists of a provided network architecture with layers $\ell_{1,...,L}$, and a fixed budget of parameters $\theta$, which are partitioned into $P$ *parameter groups* (both hyperparameters) containing parameters $\theta_{1,...,P}$. Each layer is associated with a single parameter group, which will provide the parameters used to implement it. This mapping is learned in a preliminary training step by training a specialized SSN and clustering its layer representations (Section 3.2). To implement each layer, an SSN morphs the parameters in its associated group to generate the necessary weights; this uses downsampling (Section 3.1.1) when the group has more parameters than needed, or upsampling (Section 3.1.2) when the group has fewer parameters than needed. SSNs allow any number of parameters to "shapeshift" into a network without necessitating changes to the model's loss, architecture, or hyperparameters, and the process can be applied automatically. Finally, we note that SSNs are simply one approach to NPAS. Appendices B-D contain ablation studies and discussion of variants we found to be less successful.

### 3.1 WEIGHT GENERATION

Weight generation implements a layer $\ell_i$, which requires weights $w_i$, using the fixed set of parameters in its associated parameter group $\theta_j$. (We assume the mapping between layers and parameter groups has already been established; see Section 3.2.) There are three cases to handle:

1. $|w_i| = |\theta_j|$ (exactly enough parameters): The parameters are used as-is.
2. $|w_i| < |\theta_j|$ (excess parameters): We perform parameter downsampling (Section 3.1.1).
3. $|w_i| > |\theta_j|$ (insufficient parameters): We perform parameter upsampling (Section 3.1.2).

We emphasize that, depending on how layers are mapped to parameter groups, both down- and upsampling may be required in an LB- or HB-NPAS model.

### 3.1.1 PARAMETER DOWNSAMPLING

When a parameter group $\theta_j$ provides more parameters than needed to implement a layer $\ell_i$, we perform template-based downsampling to generate $w_i$. To do this, we first split $\theta_j$ into up to $K$ (a hyperparameter) *templates* $T_i^{1,...,K}$, where each template $T_i^k$ is the same dimension as $w_i$. If $\theta_j$ does not evenly divide into templates, we ignore excess parameters. To avoid uneven sharing of parameters between layers, the templates for each layer are constructed from $\theta_j$ in a round-robin fashion. These templates are then combined to produce $w_i$; if only one template can be produced we instead use it directly. We present two different methods of learning to combine templates. To simplify presentation, we will assume there are exactly $K$ templates used.

**WAvg** (Savarese & Maire, 2019) This learns a vector $\alpha_i \in \mathbb{R}^K$ which is used to produce a weighted average of the templates: $w_i = \sum_{k=1}^{K} \alpha_i^k T_i^k$. The $\alpha_i$ are initialized orthogonally to the $\alpha$s of all other

layers in the same parameter group. While efficient, this only implicitly learns similarities between layers. Empirically, we find that different layers often converge to similar $\alpha$s, limiting sharing.

**Emb** To address this, we can instead more directly learn a representation of the layer using a *layer embedding*. We use a learnable vector $\phi_i \in \mathbb{R}^E$, where $E$ is the size of the layer representation; we use $E = 24$ throughout, as we found it to work well. A linear layer, which is shared between all layers in the parameter group and parameterized by $W_j \in \mathbb{R}^{K \times E}$ and $b_j \in \mathbb{R}^K$, is then used to construct an $\alpha_i$ for the layer, which is used as in WAvg. That is, $\alpha_i = W_j \phi_i + b_j$ and $w_i = \sum_{k=1}^{K} \alpha_i^k T_i^k$. We considered more complex methods (*e.g.*, MLPs, nonlinearities), but they did not improve performance.

While both methods require additional parameters, this is quite small in practice. WAvg requires $K$ additional parameters per layer. Emb requires $E = 24$ additional parameters per layer and $KE + K = 24K + K$ parameters per parameter group.

### 3.1.2 Parameter Upsampling

If instead a parameter group $\theta_j$ provides fewer parameters than needed to implement a layer $\ell_i$, we upsample $\theta_j$ to be the same size as $w_i$. As a layer will use all of the parameters in $\theta_j$, we do not use templates. We consider two methods for upsampling below.

**Inter** As a naïve baseline, we use bilinear interpolation to directly upsample $\theta_j$. However, this could alter the patterns captured by parameters, as it effectively stretches the receptive field. In practice, we found fully-connected and recurrent layers could compensate for this warping, but it degraded convolutional layers compared to simpler approaches such as tiling $\theta_j$.

**Mask** To address this, and avoid redundancies created by directly repeating parameters, we propose instead to use a learned mask to modify repeated parameters. For this, we first use $n = \lceil |w_i| / |\theta_j| \rceil$ tiles of $\theta_j$ to be the same size as $w_i$ (discarding excess in the last tile). We then apply a separate learned mask to each tile after the first (*i.e.*, there are $n - 1$ masks). All masks are a fixed "window" size, which we take to be 9 by default (to match the size of commonly-used $3 \times 3$ kernels in CNNs), and are shared within each parameter group. To apply, masks are multiplied element-wise over sequential windows of their respective tile. While the number of additional parameters depends on the amount of upsampling required, as the masks are small, this is negligible.

## 3.2 Mapping Layers to Parameter Groups

We now discuss how SSNs can automatically learn to assign layers to parameter groups in such a way that parameters can be efficiently shared. This is in contrast to prior work on parameter sharing (*e.g.*, Ha et al., 2016; Savarese & Maire, 2019; Jaegle et al., 2021), which required layers to be manually assigned to parameter groups. Finding an optimal mapping of layers to parameter groups is challenging, and a brute-force approach is computationally infeasible. We rely instead on SSNs learning a representation for each layer as part of the template-based parameter downsampling process, and then use this representation to identify similar layers which can effectively share parameters.

To do this, we perform a short preliminary training step in which we train a small (*i.e.*, low parameter budget) SSN version of the model using a single parameter group and a modified means of generating templates for parameter downsampling. Specifically, for a layer $\ell_i$, we split $\theta$ into $K'$ evenly-sized templates $T_i^{1,\ldots,K'}$. Since we wish to use downsampling-based weight generation, each $T_i^{k'}$ is then resized with bilinear interpolation to be the same size as $w_i$. Next, we train the SSN as usual, using WAvg or Emb downsampling with the modified templates for weight generation (there is no upsampling). By using a small parameter budget and template-based weight generation where each template comes from the same underlying parameters, we encourage significant sharing between layers so we can measure the effectiveness of sharing. We found that using a budget equal to the number of weights of the largest single layer in the network to work well. Further, this preliminary training step is short, and requires only 10–15% of the typical network training time.

Finally, we construct the parameter groups by clustering the learned layer representations into $P$ groups. As the layer representation, we take the $\alpha_i$ or $\phi_i$ learned for each layer by WAvg or Emb downsampling (resp.). We then use $k$-means clustering to group these representations into $P$ groups, which become the parameter groups used by the full SSN.

## 4  EXPERIMENTS

Our experiments include a wide variety of tasks and networks in order to demonstrate the broad applicability of NPAS and SSNs. We adapt code and data splits made available by the authors and report the average of five runs for all comparisons except ImageNet and ALBERT, which average three runs. A more detailed discussion on SSN hyperparameter settings can be found in Appendices B-D. In our paper we primarily evaluate methods based on task performance, but we demonstrate that SSNs improve reduce training time and memory in distributed learning settings in Appendix G.

**Compared Tasks.** We briefly describe each task, datasets, and evaluation metrics. For each model, we use the authors' implementation and hyperparameters, unless noted (more details in Appendix A).

**Image Classification.** For image classification the goal is to recognize if an object is present in an image. This is evaluated using Error@$k$, *i.e.*, the portion of times that the correct category does not appear in the top $k$ most likely objects. We evaluate SSNs on CIFAR-10 and CIFAR-100 (Krizhevsky, 2009), which are composed of 60K images of 10 and 100 categories, respectively, and ImageNet (Deng et al., 2009), which is composed of 1.2M images containing 1,000 categories. We report Error@1 on CIFAR and Error@5 for ImageNet.

**Image-Sentence Retrieval.** In image-sentence retrieval the goal is match across modalities (sentences and images). This task is evaluated using Recall@K={1, 5, 10} for both cross-modal directions (six numbers), which we average for simplicity. We benchmark on Flickr30K (Young et al., 2014) which contains 30K/1K/1K images for training/testing/validation, and COCO (Lin et al., 2014), which contains 123K/1K/1K images for training/testing/validation. For both datasets each image has about five descriptive captions. We evaluate SSNs using EmbNet (Wang et al., 2016) and ADAPT-T2I (Wehrmann et al., 2020). Note that ADAPT-T2I has identical parallel layers (*i.e.*, they need different outputs despite having the same input), which makes sharing parameters challenging.

**Phrase Grounding.** Given a phrase the task is to find its associated image region. Performance is measured by how often the predicted box for a phrase has at least 0.5 intersection over union with its ground truth box. We evaluate on Flickr30K Entities (Plummer et al., 2017) which augments Flickr30K with 276K bounding boxes for phrases in image captions, and ReferIt (Kazemzadeh et al., 2014), which contains 20K images that are evenly split between training/testing and 120K region descriptions. We evaluate our SSNs with SimNet (Wang et al., 2018) using the implementation from Plummer et al. (2020) that reports state-of-the-art results on this task.

**Question Answering.** For this task the goal is to answer a question about a textual passage. We use SQuAD v1.1 (Rajpurkar et al., 2016), which has 100K+ question/answer pairs on 500+ articles, and SQuAD v2.0 (Rajpurkar et al., 2018), which adds 50K unanswerable questions. We report F1 and EM scores on the development split. We compare our SSNs with ALBERT (Lan et al., 2020), a recent transformer architecture that incorporates extensive, manually-designed parameter sharing.

### 4.1  RESULTS

We begin our evaluation in low-budget (LB-NPAS) settings. Figure 3 reports results on image classification, including WRNs (Zagoruyko & Komodakis, 2016), DenseNets (Huang et al., 2017), and EfficientNets (Tan & Le, 2019); Table 1 contains results on image-sentence retrieval and phrase grounding. For each task and architecture we compare SSNs to same parameter-sized networks without sharing. In image classification, we also report results for SWRN (Savarese & Maire, 2019) sharing; but note it cannot train a WRN-28-10 or WRN-50-2 with fewer than 12M or 40M parameters, resp. We show that SSNs can create high-performing models with fewer parameters than SWRN is capable of, and actually outperform it using 25% and 60% fewer parameters on C-100 and ImageNet, resp. Table 1 demonstrates that these benefits generalize to vision-language tasks. In Table 2 we also compare SSNs with ALBERT (Lan et al., 2020), which applies manually-designed parameter sharing to BERT (Devlin et al., 2019), and find that SSN's learned parameter sharing outperforms ALBERT. This demonstrates that SSNs can implement large networks with lower memory requirements than is possible with current methods by effectively sharing parameters.

We discuss the runtime and memory performance implications of SSNs extensively in Appendix G. In short, by reducing parameter counts, SSNs reduce communication costs and memory. For example,

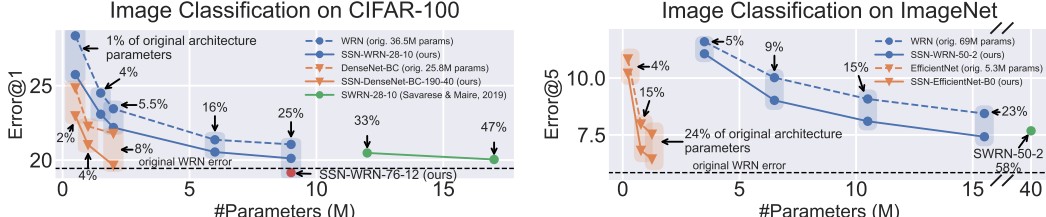

Figure 3: Comparison of reducing the parameter parameters by adjusting layer sizes vs. our SSNs using the same number of parameters. We also compare to prior work in parameter sharing, specifically SWRN (Savarese & Maire, 2019). Note that due to architecture assumptions made by SWRN, it cannot reduce the number of parameters of a WRN-28-10 by more than than 33% of the original and 58% of the parameters of a WRN-50-2. In contrast, we can not only use far fewer parameters that perform better than SWRN, but we can also implement even larger networks (WRN-76-12) without having to increase our parameter usage. See text for additional discussion.

Table 1: SSNs vs. similar size traditional networks on image-sentence retrieval and phrase grounding.

| Dataset | Method | | Performance | | |
|---|---|---|---|---|---|
| **Image-Sentence Retrieval** | | % orig params | 6.9% | 13.9% | 100% |
| F30K | EmbNet | | 56.5 | 70.2 | 74.1 |
| | SSN-EmbNet | | **67.9** | **71.6** | **74.4** |
| COCO | EmbNet | | 74.2 | 77.3 | 81.4 |
| | SSN-EmbNet | | **76.7** | **78.2** | **82.1** |
| | | % orig params | 34.7% | 70.9% | 100% |
| F30K | ADAPT-T2I | | 76.6 | 80.6 | **83.3** |
| | SSN-ADAPT-T2I | | **81.2** | **81.7** | 82.9 |
| COCO | ADAPT-T2I | | 83.8 | 85.2 | 86.8 |
| | SSN-ADAPT-T2I | | **85.3** | **86.2** | **87.0** |
| **Phrase Grounding** | | % orig params | 7.8% | 15.6% | 100% |
| F30K Ent | SimNet | | 70.6 | 71.0 | 71.7 |
| | SSN-SimNet | | **71.9** | **72.1** | **72.4** |
| ReferIt | SimNet | | 58.5 | 59.2 | **61.1** |
| | SSN-SimNet | | **59.5** | **59.9** | 61.0 |

our SSN-ALBERT-Large trains about $1.4\times$ faster using 128 GPUs than BERT-Large (in line with results for ALBERT), and reduces memory requirements by about 5 GB (1/3 of total).

As mentioned before, knowledge distillation and parameter pruning can help create more efficient models at test time, although they cannot reduce memory requirements during training like SSNs. Tables 3 and 4 show our approach can be used to accomplish a similar goal as these tasks. Comparing our LB-NPAS results in Table 4 and the lowest parameter setting of HRank, we report a 1.5% gain over pruning methods even when using less than half the parameters. We note that one can think of our SSNs in the high budget setting (HP-NPAS) as mixing together a set of random initializations of a network by learning to combine the different templates. This setting's benefit is illustrated in Table 3 and Table 4, where our HB-NPAS models report a 1-1.5% gain over training a traditional network. As a reminder, in this setting we precompute the weights of each layer once training is complete so they require no additional overhead at test time. That said, best performance on both tasks comes from combining our SSNs with prior work on both tasks.

## 4.2 ANALYSIS OF SHAPESHIFTER NETWORKS

In this section we present ablations of the primary components of our approach. A complete ablation study, including, but not limited to comparing the number of parameter groups (Section 3.2) and number of templates ($K$ in Section 3.1.1) can be found in Appendices B-D.

Table 5 compares the strategies generating weights from templates described in Section 3.1.1 when using a single parameter group. In these experiments we set the number of parameters as the amount required to implement the largest layer in a network. For example, the ADAPT-T2I model requires 14M parameters, but its bidirectional GRU accounts for 10M of those parameters, so all SSN variants in this experiment allocate 10M parameters. Comparing to the baseline, which involves modifying the original model's number and/or size of filters so they have the same number of parameters as our

Table 2: Comparison of ALBERT-base and -large (Lan et al., 2020) vs SSNs using $P = 4$ learned parameter groups. Scores are F1 and EM on dev sets.

| Network | SQuAD v1.1 | SQuAD v2.0 |
|---------|-----------|-----------|
| Base | 12M params (11% of orig) | |
| ALBERT | 89.2/82.1 | 79.8/76.9 |
| LB-SSN | **90.1/83.0** | **80.7/78.0** |
| Large | 18M params (5% of orig) | |
| ALBERT | 90.5/83.7 | 82.1/79.3 |
| LB-SSN | **91.1/84.4** | **83.0/80.1** |

Table 3: Knowledge distillation experiments on CIFAR-100 using OFD (Heo et al., 2019) and SRR (Yang et al., 2021). HB-SSN's budget is $4\times$ the student's size. For all settings using our SSNs improve performance over distillation alone.

| Teacher | Error@1 | Student | Baseline | OFD | SRR |
|---------|---------|---------|----------|-----|-----|
| WRN-40-4 | 20.50 | WRN-16-2 | 27.30 | 24.43 | 24.03 |
| | | HB-SSN | **26.53** | **23.99** | **23.42** |
| ResNet-34 | 27.95 | ResNet-10 | 31.58 | 31.06 | 30.09 |
| | | HB-SSN | **30.88** | **30.41** | **29.63** |

Table 4: Parameter pruning experiments with HRank (Lin et al., 2020). HB-SSN's budget is $4\times$ the model's size during pretraining, then we generate the final weights for pruning. In all settings using our SSNs improve performance over pruning alone.

| Method | C-10 Error@1 | | % original test FLOPs | % original test params |
|--------|-----------|----------|------------|-------------|
| | ResNet-56 | HB-SSN | | |
| Base | 6.74 | **5.21** | 100.0 | 100.0 |
| HRank | 6.48 | **5.86** | 70.7 | 83.2 |
| HRank | 10.75 | **9.94** | 20.3 | 28.4 |
| LB-SSN | 9.26 | – | 100.6 | 12.2 |

Table 5: Parameter downsampling comparison (Section 3.1.1) using WRN-28-10 and WRN-50-2 for C-10/100 and ImageNet, resp. Baseline adjusts the number and/or size of filters rather than share parameters. See Appendix B for additional details.

| Dataset | % orig params | Reduced Baseline | SSNs (ours) | |
|---------|---------|----------|------|-----|
| | | | WAvg | Emb |
| C-10 | 11.3% | 4.22 | 4.00 | **3.84** |
| C-100 | | 22.34 | **21.78** | 21.92 |
| ImageNet | 27.5% | 10.08 | 7.38 | **6.69** |

SSNs, we see that the variants of our SSNs perform better, especially on ImageNet where we reduce Error@5 by 3%. Generally we see a slight boost to performance using Emb over WAvg. Also note that prior work in parameter sharing, *e.g.*, SWRN (Savarese & Maire, 2019), can not be applied to the settings in Table 5, since they require parameter sharing between layers of different types and different operations, or have too few parameters. *E.g.*, the WRN-28-10 results use just under 4M parameters, but, as shown in Figure 3(a), SWRN requires a minimum of 12M parameters.

In Table 6 we investigate one of the new challenges in this work: how to upsample parameters so a large layer operation can be implemented with relatively few parameters (Section 3.1.2). For example, our SSN-WRN-28-10 results use about 0.5M parameters, but the largest layer defined in the network requires just under 4M weights. We find using our simple learned Mask upsampling method performs well in most settings, especially when using convolutional networks. For example, on CIFAR-100 it improves Error@1 by 2.5% over the baseline, and 1.5% over using bilinear interpolation (inter). While more complex methods of upsampling may seem like they would improve performance (*e.g.*, using an MLP, or learning combinations of basis filters), we found such approaches had two significant drawbacks. First, they can slow down training time significantly due to their complexity, so only a limited number of settings are viable. Second, in our experiments we found many had numerical stability issues for some datasets/tasks. We believe this may be due to trying to learn the local parameter patterns and the weights to combine them concurrently. Related work suggests this can be resolved by leveraging prior knowledge about what these local parameter patterns should represent (Denil et al., 2013), *i.e.*, you define and freeze what they represent. However, prior knowledge is not available in the general case, and data-driven methods of training these local filter patterns often rely on pretraining steps of the fully-parameterized network (*e.g.*, Denil et al., 2013). Thus, they are not suited for NPAS since they can not training large networks with low memory requirements, but addressing this issue would be a good direction for future work.

Table 7 compares approaches for mapping layers to parameter groups using the same number of parameters as the original model. We see a small, but largely consistent improvement over using a traditional (baseline) network using SSNs. Notably, our automatically learned mappings (auto) perform on par with manual groups. This demonstrates that our automated approach can be used without loss in performance, while being applicable to *any* architecture, making them more flexible than hand-crafted methods. This flexibility does come with a computational cost, as our preliminary step that learns to map layers to parameter groups resulted in a 10-15% longer training time for equivalent epochs. That said, parameter sharing methods have demonstrated an ability to converge

Figure 4: Qualitative comparison of the **(a)** generated weights for a WRN-16-2 and **(b)** the parameter groups for a WRN-28-10 model trained on C-100. See Section 4.2 for discussion.

Table 6: Parameter upsampling comparison (Section 3.1.2). "Reduced Baseline" adjusts the number and/or size of filters rather than share parameters. See Appendix B for the original models' performance.

| Dataset | % orig params | Reduced Baseline | SSNs (ours) Inter | SSNs (ours) Mask |
|---|---|---|---|---|
| C-10 | 1.4% | 6.35 | 6.05 | **5.07** |
| C-100 | | 28.35 | 26.84 | **25.20** |
| ImageNet | 2.2% | 15.20 | 14.83 | **14.19** |

Table 7: Compares methods of creating parameter groups (Section 3.2). These results demonstrate that our automatically learned mappings (auto) provide more flexibility without reducing performance compared to manual mappings.

| Dataset | Baseline | SSNs (ours) Single | SSNs (ours) Random | SSNs (ours) Manual | SSNs (ours) Auto |
|---|---|---|---|---|---|
| C-10 | 3.57 | 3.71 | 3.63 | **3.38** | 3.42 |
| C-100 | 19.44 | 19.99 | 20.36 | 19.29 | **19.24** |
| ImageNet | 5.84 | 6.18 | 5.91 | **5.82** | 5.86 |

faster (Bagherinezhad et al., 2017; Lan et al., 2020). Thus, exploring more efficient training strategies using NPAS methods like SSNs will be a good direction for future work.

Figure 4(a) compares the $3 \times 3$ kernel filters at the early, middle, and final convolutional layers of a WRN-16-2 of a traditional neural network (no parameter sharing) and our SSNs where all layers belong to the same parameter group. We observe a correspondence between filters in the early layers, but this diverges in deeper layers. This suggests that sharing becomes more difficult in the final layers, which is consistent with two observations we made about Figure 4(b), which visualizes parameter groups used for SSN-WRN-28-10 to create 14 parameter group mappings. First, we found the learned parameter group mappings tended to share parameters between layers early in the network, opting for later layers to share no parameters. Second, the early layers tended to group layers into 3–4 parameter stores across different runs, with the remaining 10–11 parameter stores each containing a single layer. Note that these observations were consistent across different random initializations.

## 5 CONCLUSION

We propose NPAS, a novel task in which the goal is to implement a given, arbitrary network architecture given a fixed parameter budget. This involves identifying how to assign parameters to layers and implementing layers with their assigned parameters (which may be of any size). To address NPAS, we introduce SSNs, which automatically learn how to share parameters. SSNs benefit from parameter sharing in the low-budget regime—reducing memory and communication requirements when training—and enable a novel high-budget regime that can improve model performance. We show that SSNs boost results on ImageNet by 3% improvement in Error@5 over a same-sized network without parameter sharing. Surprisingly, we also find that parameters can be shared among very different layers. Further, we show that SSNs can be combined with knowledge distillation and parameter pruning to achieve state-of-the-art results that also reduce FLOPs at test time. One could think of SSNs as spreading the same number of parameters across more layers, increasing effective depth, which benefits generalization (Telgarsky, 2016), although this requires further exploration.

**Acknowledgements.** This work is funded in part by grants from the National Science Foundation and DARPA. This project received funding from the European Research Council (ERC) under the European Union's Horizon 2020 program (grant agreement MAELSTROM, No. 955513). N.D. is supported by the ETH Postdoctoral Fellowship. We thank the Livermore Computing facility for the use of their GPUs for some experiments.

## ETHICS STATEMENT

Neural Parameter Allocation Search and Shapeshifter Networks are broad and general-purpose, and therefore applicable to many downstream tasks. It is thus challenging to identify specific cases of benefit or harm, but we note that reducing memory requirements can have broad implications. In particular, this could allow potentially harmful applications to be cheaply and widely deployed (*e.g.*, facial recognition, surveillance) where it would otherwise be technically or economically infeasible.

## REPRODUCIBILITY STATEMENT

NPAS is a task which can be implemented in many different ways; we define it formally in Section 2. SSNs, our proposed solution to NPAS, are presented in detail in Section 3, and Figure 2 provides an illustration and example for weight generation methods. AppendixA also provides a thorough discussion of the implementation details. To further aid reproducibility, we publicly release our SSN code at `https://github.com/BryanPlummer/SSN`.

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

# A    DESCRIPTION OF COMPARED TASKS

## A.1    IMAGE-SENTENCE RETRIEVAL

In bidirectional image-sentence retrieval when a model is provided with an image the goal is to retrieve a relevant sentence and vice versa. This task is evaluated using Recall@K={1, 5, 10} for both directions (resulting in 6 numbers), which we average for simplicity. We benchmark methods on two common datasets: Flickr30K (Young et al., 2014) which contains 30K/1K/1K images for training/testing/validation, each with five descriptive captions, and MSCOCO (Lin et al., 2014), which contains 123K/1K/1K images for training/testing/validation, each image having roughly five descriptive captions.

**EmbNet (Wang et al., 2016).** This network learns to embed visual features for each image computed using a 152-layer Deep Residual Network (ResNet) (He et al., 2016) that has been trained on ImageNet (Deng et al., 2009) and the average of MT GrOVLE (Burns et al., 2019) language features representing each word into a shared semantic space using a triplet loss. The network consists of two branches, one for each modality, and each branch contains two fully connected layers (for a total of four layers). We adapted the implementation of Burns *et al*. (Burns et al., 2019)[1], and left all hyperparameters at the default settings. Specifically, we train using a batch size of 500 with an initial learning rate of 1e-4 which we decay exponentially with a gamma of 0.794 and use weight decay of 0.001. The model is trained until it has not improved performance on the validation set over the last 10 epochs. This architectures provides a simple baseline for parameter sharing with our Shapeshifter Networks (SSNs), where layers operate on two different modalities.

**ADAPT-T2I (Wehrmann et al., 2020).** In this approach word embeddings are aggregated using a bidirectional GRU (Cho et al., 2014) and its hidden state at each timestep is averaged to obtain a full-sentence representation. Images are represented using 36 bottom-up image region features (Anderson et al., 2018) that are passed through a fully connected layer. Then, each sentence calculates scaling and shifting parameters for the image regions using a pair of fully connected layers that both take the full-sentence representation as input. The image regions are then averaged, and a similarity score is computed between the sentence-adapted image features and the fully sentence representation. Thus, there are four layers total (3 fully connected, 1 GRU) that can share parameters, including the two parallel fully connected layers (*i.e.*, they both take the full sentence features as input, but are expected to have different outputs). We adapted the author's implementation and kept the default hyperparameters[2]. Specifically, we use a latent dimension of 1024 for our features and train with a batch size of 105 using a learning rate of 0.001. This method was selected since it achieves high performance and also included fully connected and recurrent layers, as well as having a set of parallel layers that make effectively performing cross-layer parameter sharing more challenging.

## A.2    PHRASE GROUNDING

Given a phrase the goal of a phrase grounding model is to identify the image region described by the phrase. Success is achieved if the predicted box has at least 0.5 intersection over union with the ground truth box. Performance is measured using the percent of the time a phrase is accurately localized. We evaluate on two datasets: Flickr30K Entities (Plummer et al., 2017) which augments the Flickr30K dataset with 276K bounding boxes associated with phrases in the descriptive captions, and ReferIt (Kazemzadeh et al., 2014) which contains 20K images that are evenly split between training/testing and 120K region descriptions.

**SimNet (Wang et al., 2018).** This network contains three branches that each operate on different types of features. One branch passes image regions features computed with a 101-layer ResNet that have been fine-tuned for phrase grounding using two fully connected layers. A second branch passes MT GrOVLE features through two fully connected layers. Then, a joint representation is computed for all region-phrase pairs using an elementwise product. Finally, the third branch passes these joint features through three fully connected layers (7 total), where the final layer acts as a classifier indicating the likelihood that phrase is present in the image region. We adapt the code

---

[1] https://github.com/BryanPlummer/Two_branch_network
[2] https://github.com/jwehrmann/retrieval.pytorch

from Plummer et al. (2020)[3] and keep all hyperameters at their default settings. Specifically, we use a pretrained Faster R-CNN model (Ren et al., 2015) fine-tuned for phrase grounding by Plummer et al. (2020) on each dataset to extract region features. Then we encode each phrase by averaging MT GrOVLE features (Burns et al., 2019) and provide the image and phrase features as input to our model. We train our model using a learning rate of 5e-5 and a final embedding dimension of 256 until it no longer improves on the validation set for 5 epochs (typically resulting in training times of 15-20 epochs). Performing experiments on this model enables us to test how well our SSNs generalize to another task and how well it can adapt to sharing parameters with layers operating on three types of features (just vision, just language, and a joint representation).

## A.3 IMAGE CLASSIFICATION

For image classification the goal is to be able to recognize if an object is present in an image. Typically this task is evaluated using Error@K, or the portion of times that the correct category doesn't appear in the top $k$ most likely objects. We evaluate our Shapeshifter Networks on three datasets: CIFAR-10 and CIFAR-100 (Krizhevsky, 2009), which are comprised of 60K images of 10 and 100 categories, respectively, and ImageNet (Deng et al., 2009), which is comprised of 1.2M images containing 1,000 categories. We report Error@1 for both CIFAR datasets and Error@5 for ImageNet. In these appendices, we also report Error@1 for ImageNet.

**Wide Residual Network (WRN) (Zagoruyko & Komodakis, 2016).** WRN modified the traditional ResNets by increasing the width $k$ of each layer while also decreasing the depth $d$, which they found improved performance. Different variants are identified using WRN-$d$-$k$. Following Savarese *et al.* (Savarese & Maire, 2019), we evaluate our Shapeshifter Networks using WRN-28-10 for CIFAR and WRN-50-2 for ImageNet. We adapt the implementation of Savarese *et al.*[4] and use cutout (De-Vries & Taylor, 2017) for data augmentation. Specifically, on CIFAR we train our model using a batch size of 128 for 200 epochs with weight decay set at 5e-4 and an initial learning rate of 0.1 which we decay using a gamma of 0.2 at 60, 120, and 160 epochs. Unlike the vision-language models discussed earlier, these architecture include convolutional layers in addition to a fully connected layer used to implement a classifier, and also have many more layers than the shallow vision-language models.

**DenseNet (Huang et al., 2017).** Unlike traditional neural networks where each layer in the network is computed in sequence, every layer in a DenseNet using feature maps from every layer which came before it. We adapt PyTorch's official implementation[5] using the hyperparameters as set in Huang *et al.* (Huang et al., 2017). Specifically, on CIFAR we train our model using a batch size of 96 for 300 epochs with weight decay set at 1e-4 and an initial learning rate of 0.1 which we decay using a gamma of 0.1 at 150 and 225. These networks provide insight into the effect depth has on learning SSNs, as we use a 190-layer DenseNet-BC configuration for CIFAR. However, due to their high computational cost we provide limited results testing only some settings.

**EfficientNet (Tan & Le, 2019).** EfficientNets are a class of model designed to balance depth, width, and input resolution in order to produce very parameter-efficient models. For ImageNet, we adapt an existing PyTorch implementation and its hyperparameters[6], which are derived from the official TensorFlow version. We use the EfficientNet-B0 architecture to illustrate the impact of SSNs on very parameter-efficient, state-of-the-art models. On CIFAR-100 we use an EfficientNet with Network Deconvolution (ND) (Ye et al., 2020), which results in improved results with similar numbers of epochs for training. We use the author's implementation[7], and train each model for 100 epochs (their best performing setting). Note that our best error running different configurations of their model (35.88) is better than those in their paper (37.63), so despite the relatively low performance it is comparable to results from their paper.

---

[3] https://github.com/BryanPlummer/phrase_detection
[4] https://github.com/lolemacs/soft-sharing
[5] https://pytorch.org/hub/pytorch_vision_densenet/
[6] https://rwightman.github.io/pytorch-image-models/
[7] https://github.com/yechengxi/deconvolution

### A.4    QUESTION ANSWERING

In question answering, a model is given a question and an associated textual passage which may contain the answer, and the goal is to predict the span of text in the passage that contains the answer. We use two versions of the Stanford Question Answering Dataset (SQuAD), SQuAD v1.1 (Rajpurkar et al., 2016), which contains 100K+ question/answer pairs on 500+ Wikipedia particles, and SQuAD v2.0, which augments SQuAD v1.1 with 50K unanswerable questions designed adversarially to be similar to standard SQuAD questions. For both datasets, we report both the F1 score, which captures the precision and recall of the chosen text span, and the Exact Match (EM) score.

**ALBERT (Lan et al., 2020)** ALBERT is a version of the BERT (Devlin et al., 2019) transformer architecture that applies cross-layer parameter sharing. Specifically, the parameters for all components of a transformer layer are shared among all the transformer layers in the network. ALBERT also includes a factorized embedding to further reduce parameters. We follow the methodology of BERT and ALBERT for reporting results on SQuAD, and our baseline ALBERT scores closely match those reported in the original work. This illustrates the ability of NPAS and SSNs to develop better parameter sharing methods than manually-designed systems for extremely large models.

## B    EXTENDED RESULTS WITH ADDITIONAL BASELINES

Below we provide additional results with more baseline methods for the three components of our SSNs: weight generator (Section B.1), parameter upsampling (Section B.4), and mapping layers to parameter groups (Section B.3). We provide ablations on the number of parameter groups and templates used by our SSNs in Section C and Section D, respectively.

### B.1    ADDITIONAL METHODS THAT GENERATE LAYER WEIGHTS FROM TEMPLATES

Parameter downsampling uses the selected templates $T_i^k$ for a layer $\ell_i$ to produce its weights $w_i$. In Section 3.1.1 of the paper we discuss two methods of learning a combination of the $T_i^k$ to generate $w_i$. Below in Section B.2 we provide two simple baseline methods that directly use the candidates. Table 8 compares the baselines to the methods in the main paper that learn weighted combinations of templates, where the learned methods typically perform better than the baselines.

### B.2    DIRECT TEMPLATE COMBINATION

Here we describe the strategies we employ that require no parameters to be learned by weight generator, *i.e.*, they operate directly on the templates $T_i^k$.

**Round Robin (RR)** reuses parameters of each template set as few times as possible. The scheme simply returns the weights at index $k \mod K$ in the (ordered) template set $T_i$ at the $k$th query of a parameter group.

**Candidate averaging (Avg)** averages all candidates in $T_i$ to provide a naive baseline for using multiple candidates. A significant drawback of this approach is that, if $K$ is large, this can result in reusing parameters (across combiners) many times with no way to adapt to a specific layer, especially when the size of the parameter group is small.

### B.3    ADDITIONAL PARAMETER MAPPING RESULTS

Table 9 compares approaches that map layers to parameter groups using the same number of parameters as the original model. We see a small, but largely consistent improvement over using a traditional (baseline) network. Notably, our learned grouping methods (WAvg, Emb) perform on par, and sometimes better than using manual mappings. However, our approach can be applied to *any* architecture to create a selected number of parameter groups, making them more flexible than hand-crafted methods. For example, in Table 10, we see using two groups often helps to improve performance when using very few parameters, but it is not clear how to effectively create two groups by hand for many networks.

Table 8: **Weight generation method comparison.** See Section B.1 for a description of the baseline methods (RR, Avg), and Section 3.1.1 of the paper for the learned candidates (WAvg, Emb). All models in the same row use the same (reduced) number of parameters. "Reduced Baseline" uses no parameter sharing, but adjusts the number and/or size of filters so they have the same number of parameters as our SSNs. The original model's performance is reported in the first column of results in Table 9.

| Method | Orig Num Params (M) | % of Params Used | Dataset | Reduced Baseline | SSN Variants (ours) | | | |
|---|---|---|---|---|---|---|---|---|
| | | | | | RR | Avg | WAvg | Emb |
| **Image-Sentence Retrieval** | | | | | (↑ better) | | | |
| EmbNet | 7 | 57.1% | F30K | 72.8 | 74.1 | 73.5 | 74.0 | **74.3** |
| | | | COCO | 80.9 | 80.8 | 80.9 | 81.2 | **81.5** |
| ADAPT-T2I | 14 | 70.9% | F30K | 80.6 | 81.2 | 80.5 | 81.6 | **81.7** |
| | | | COCO | 85.2 | 85.6 | 85.8 | **86.2** | 85.9 |
| **Phrase Grounding** | | | | | (↑ better) | | | |
| SimNet | 6 | 33.1% | F30K Entities | 71.1 | 72.3 | 72.1 | 72.3 | **72.5** |
| | | | ReferIt | 59.4 | **60.5** | 60.2 | **60.5** | 60.4 |
| **Image Classification** | | | | | (↓ better) | | | |
| WRN-28-10 | 36 | 11.3% | C-10 | 4.22 | 4.09 | 4.19 | 4.00 | **3.84** |
| | | | C-100 | 22.34 | 21.91 | 22.78 | **21.78** | 21.92 |
| WRN-50-2 | 69 | 27.5% | ImageNet E@5 | 10.08 | **6.69** | 7.61 | 7.38 | **6.69** |
| | | | ImageNet E@1 | 28.68 | 23.32 | 25.11 | 24.15 | **23.25** |

Table 9: **Parameter mapping comparison** (described in Section 3.2 of the paper). Each method uses the same number of parameters and the baseline represents no parameter sharing. Notably, these results demonstrate how our automatically learned groups can provide greater flexibility when sharing parameters while performing on par or better than manually created groups (which are illustrated in Figure 5).

| Method | #Groups | Dataset | Baseline | SSN Variants (ours) | | | | |
|---|---|---|---|---|---|---|---|---|
| | | | | Single | Random | Manual | WAvg | Emb |
| **Image-Sentence Retrieval** | | | | (↑ better) | | | | |
| EmbNet | 2 | F30K | 74.1 | **74.4** | 74.0 | 74.3 | 74.2 | 74.3 |
| | | COCO | 81.4 | **82.1** | 81.5 | 81.7 | 81.7 | 81.9 |
| ADAPT-T2I | 2 | F30K | **83.3** | 82.1 | 81.9 | 82.0 | 82.6 | 82.9 |
| | | COCO | 86.8 | 86.1 | 86.3 | 86.1 | 86.4 | **87.0** |
| **Phrase Grounding** | | | | (↑ better) | | | | |
| SimNet | 3 | F30K Entities | 71.7 | 71.4 | 71.8 | **72.4** | 72.2 | 72.1 |
| | | ReferIt | **61.1** | 60.9 | 60.0 | 60.2 | 61.0 | 60.5 |
| **Image Classification** | | | | (↓ better) | | | | |
| WRN-28-10 | 14 | C-10 | 3.57 | 3.71 | 3.63 | **3.38** | 3.51 | 3.42 |
| | | C-100 | 19.44 | 19.99 | 20.36 | 19.29 | 19.47 | **19.24** |
| WRN-50-2 | 32 | ImageNet E@5 | 5.84 | 6.18 | 5.91 | **5.82** | 5.86 | 5.96 |
| | | ImageNet E@1 | 21.50 | 22.09 | 21.54 | **21.43** | 21.59 | 21.61 |

## B.4 EXTENDED PARAMETER UPSAMPLING

In Table 10 we provide extended results comparing the parameter upsamping methods. We additionally compare with a further naïve baseline of simply repeating parameters until they are the appropriate size. We find that Mask upsampling is always competitive, and typically moreso when two parameter groups are used.

## B.5 COMPARISON WITH HYPERNETWORKS

In Table 11 we compare our SSNs on Wide ResNets (Ha et al., 2016) to the same networks implemented using Hypernetworks (Ha et al., 2016) for CIFAR-10, using the results reported in their paper. We can see that, for the same parameter budget, SSNs outperform Hypernetworks.

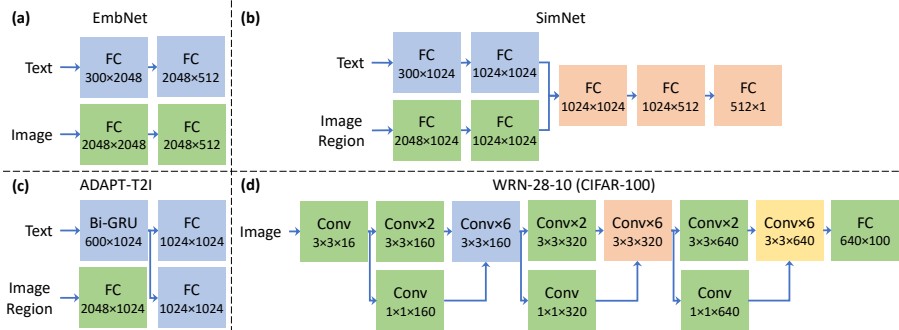

Figure 5: Examples of architectures used to evaluate SSNs. Fully connected (FC) and bidirectional GRU (Bi-GRU) layers give their $in \times out$ dimensions; convolutional layers (conv) give the number of layers, kernel size, and number of filters. Same colored boxes show "manual" (Table 9) sharing, except for green boxes in WRN, whose layers share no parameters. Layers without parameters omitted.

Table 10: **Parameter upsampling comparison.** Most methods are described in Section 3.1.2 of the paper, except the method "repeat" that simply duplicates weights when upsampling. "Reduced Baseline" uses no parameter sharing, but adjusts the number and/or size of filters so they have the same number of parameters as our SSNs. The original model's performance is reported in the first column of results in Table 9.

| Method | Orig Num Params (M) | % of Params Used | Dataset | Reduced Baseline | SSN Variants (ours) | | | |
|---|---|---|---|---|---|---|---|---|
| | | | | | Repeat | Inter | Mask | 2 Groups+Mask |
| **Image-Sentence Retrieval** | | | | | (↑ better) | | | |
| EmbNet | 6 | 6.9% | F30K | 56.5 | 64.8 | 67.5 | 67.3 | **67.9** |
| | | | COCO | 74.2 | 75.5 | 76.1 | 76.2 | **76.7** |
| ADAPT-T2I | 14 | 34.7% | F30K | 76.6 | 79.0 | 80.9 | 80.3 | **81.2** |
| | | | COCO | 83.8 | 84.9 | **85.3** | 84.6 | 84.9 |
| **Phrase Grounding** | | | | | (↑ better) | | | |
| SimNet | 6 | 7.8% | F30K Entities | 70.6 | 71.1 | 71.8 | 71.5 | **71.9** |
| | | | ReferIt | 58.5 | 58.9 | 59.3 | 59.2 | **59.5** |
| **Image Classification** | | | | | (↓ better) | | | |
| WRN-28-10 | 36 | 1.4% | C-10 | 6.35 | 5.43 | 6.05 | 5.18 | **5.07** |
| | | | C-100 | 28.35 | 27.08 | 26.84 | 25.71 | **25.20** |
| WRN-50-2 | 69 | 2.2% | ImageNet E@5 | 15.20 | **14.16** | 14.83 | 14.19 | 14.58 |
| | | | ImageNet E@1 | 37.05 | **35.48** | 36.60 | 35.72 | 36.01 |

## C  EFFECT OF THE NUMBER OF PARAMETER GROUPS $P$

A significant advantage of using learned mappings of layers to parameter groups, described in Section 3.2, is that our approach can support any number of parameter groups, unlike prior work that required manual grouping and/or heuristics to determine which layers shared parameters (*e.g.*, Lan et al., 2020; Savarese & Maire, 2019). In this section we explore how the number of parameter groups

Table 11: Comparison of SSNs to Hypernetworks (Ha et al., 2016) on CIFAR-10.

| Method | Error@1 | #Params (M) |
|---|---|---|
| Wide ResNet 40-1 | 6.73 | 0.563 |
| Hyper Wide ResNet | 8.02 | 0.097 |
| SSN (ours) | **7.22** | 0.097 |
| Wide ResNet 40-2 | 5.66 | 2.236 |
| Hyper Wide ResNet | 7.23 | 0.148 |
| SSN (ours) | **6.56** | 0.148 |

Table 12: Effect the number of learned parameter groups $P$ has on performance for a SSN-WRN-28-10 model when training on C-100 using the Emb strategy. See Section C for discussion.

| #Groups | 1 | 2 | 4 | 8 | 12 | 14 | 16 |
|---|---|---|---|---|---|---|---|
| 4M Params | $21.92 \pm 0.30$ | $21.47 \pm 0.23$ | $\mathbf{21.31 \pm 0.13}$ | $22.01 \pm 0.26$ | $22.43 \pm 0.37$ | $23.88 \pm 0.40$ | $24.99 \pm 0.39$ |
| 36M Params | $19.99 \pm 0.22$ | $19.80 \pm 0.38$ | $19.75 \pm 0.29$ | $19.69 \pm 0.28$ | $19.27 \pm 0.10$ | $\mathbf{19.24 \pm 0.29}$ | $19.40 \pm 0.14$ |

Table 13: Effect the number of templates $K$ has on performance. Share type is set using the best results from the main paper for each method. Due to the computational cost, we set the number of candidates for ImageNet experiments in our paper using CIFAR results. See Section D for discussion

| Method | #Candidates | 2 | 4 | 6 | 8 | 10 |
|---|---|---|---|---|---|---|
| **Image-Sentence Retrieval** | | | | | | |
| EmbNet | F30K | $73.7 \pm 0.55$ | $\mathbf{74.3 \pm 0.25}$ | $73.9 \pm 0.18$ | $\mathbf{74.3 \pm 0.14}$ | $74.1 \pm 0.16$ |
| | COCO | $81.7 \pm 0.26$ | $\mathbf{81.9 \pm 0.30}$ | $81.6 \pm 0.23$ | $81.7 \pm 0.16$ | $81.5 \pm 0.19$ |
| **Phrase Grounding** | | | | | | |
| SimNet | F30K Entities | $72.1 \pm 0.18$ | $\mathbf{72.2 \pm 0.34}$ | $72.0 \pm 0.33$ | $71.6 \pm 0.29$ | $71.8 \pm 0.25$ |
| | ReferIt | $60.7 \pm 0.37$ | $\mathbf{61.0 \pm 0.43}$ | $60.8 \pm 0.39$ | $60.4 \pm 0.45$ | $60.5 \pm 0.43$ |
| **Image Classification** | | | | | | |
| WRN-28-10 C-100 | | $19.66 \pm 0.23$ | $19.59 \pm 0.21$ | $19.48 \pm 0.24$ | $\mathbf{19.24 \pm 0.29}$ | $19.32 \pm 0.27$ |

effects performance on the image classification task. We do not benchmark bidirectional retrieval and phrase grounding since networks addressing these tasks have few layers, so parameter groups are less important (as shown in Table 7).

Table 12 reports the performance of our SSNs when using different numbers $P$ parameter groups. We find that when training with few parameters (first line) low numbers of parameter groups work best, while when more parameters are available larger numbers of groups work better (second line). In fact, there is a significant drop in performance going from 4 to 8 groups when training with few parameters as seen in the first line of Table 12. This is due to the fact that starting at 8 groups some parameter groups had too few weights to implement their layers, resulting in extensive parameter upsampling. This suggests that we may be able to further improve performance when there are few parameters by developing better methods of implementing layers when too few parameters are available.

## D    EFFECT OF THE NUMBER OF TEMPLATES $K$

Table 13 reports the results using different numbers of templates. We find that varying the number of templates only has a minor impact on performance most of the time. We note that more templstes tends to lead to reduced variability between runs, making results more stable. As a reminder, however, the number of templates does not guarantee that each layer will have enough parameters to construct them. Thus, parameter groups only use this hyperparameter when many weights are available to it (*i.e.*, it can form multiple templates for the layers it implements). This occurs for the phrase grounding and bidirectional retrieval results at the higher maximum numbers of templates.

## E    SCALING SSNS TO LARGER NETWORKS

Table 14 demonstrates the ability of our SSNs to significantly reduce the parameters required, and thus the memory required to implement large Wide ResNets so they fall within specific bounds. For example, Table 14(b) shows larger and deeper configurations continue to improve performance even when the number of parameters remains largely constant. Comparing the first line of Table 14(a) and the last line of Table 14(c) we see that SSN-WRN-76-12 outperforms the fully-parameterized WRN-28-10 network by 0.6% on CIFAR-100 while only using just over half the parameters, and comes within 0.5% of WRN-76-12 while only using 13.0% of its parameters. We do note that using a SSN does not reduce the number of floating point operations, so although our SSN-WRN-76-12 model uses fewer parameters than the WRN-28-10, it is still slower at both test and train time. However, our results help demonstrate that SSNs can be used to implement very large networks with lower memory

Table 14: Comparison of different Wide Resnet (Zagoruyko & Komodakis, 2016) configurations under different Shapeshifter Network settings on the image classification task. **(a)** contains baseline results without parameter sharing, while **(b)** demonstrates that using SSNs enable us to improve performance by increasing the architecture width or depth without also increasing memory consumed, and **(c)** shows that the gap between SSNs with few model weights and fully-parameterized baselines in **(a)** gets smaller as the networks get larger. Note that the number of parameters in **(c)** was set by summing together the number of parameters required to implement the largest layer in the learned parameter group mappings for each network (*e.g.*, four parameter groups would result in summing four numbers). See Section E for discussion.

| | Method | Orig Num Params (M) | #Params Used (M) | % of Orig Params | % of WRN-28-10 Params | CIFAR-100 Error@1 |
|---|---|---|---|---|---|---|
| **(a)** | WRN-28-10 | 36 | 36 | 100.0 | 100.0 | $19.44 \pm 0.21$ |
| | WRN-40-10 | 56 | 56 | 100.0 | 153.1 | $18.73 \pm 0.34$ |
| | WRN-52-10 | 75 | 75 | 100.0 | 206.2 | $18.85 \pm 0.31$ |
| | WRN-76-12 | 164 | 164 | 100.0 | 449.6 | $\mathbf{18.31 \pm 0.22}$ |
| **(b)** | SSN-WRN-28-10 | 36 | 4 | 11.3 | 11.3 | $21.71 \pm 0.13$ |
| | SSN-WRN-40-10 | 56 | 4 | 6.8 | 10.3 | $21.06 \pm 0.11$ |
| | SSN-WRN-52-10 | 75 | 4 | 5.0 | 10.4 | $20.93 \pm 0.25$ |
| | SSN-WRN-76-12 | 164 | 5 | 3.3 | 14.9 | $\mathbf{20.37 \pm 0.21}$ |
| **(c)** | SSN-WRN-28-10 | 36 | 4 | 10.3 | 10.3 | $21.78 \pm 0.16$ |
| | SSN-WRN-40-10 | 56 | 7 | 13.4 | 20.4 | $20.17 \pm 0.32$ |
| | SSN-WRN-52-10 | 75 | 15 | 19.7 | 40.7 | $19.50 \pm 0.21$ |
| | SSN-WRN-76-12 | 164 | 21 | 13.0 | 58.6 | $\mathbf{18.83 \pm 0.19}$ |

Table 15: Performance of SSNs vs. traditional networks of a comparable size on image classification on CIFAR-100 (from Figure 3 of the main paper). See Section 4.1 for discussion.

| Method | Performance | | | | |
|---|---|---|---|---|---|
| % orig params | 1.4% | 4.1% | 5.5% | 16.4% | 24.6% |
| WRN (Huang et al., 2017) | 28.35 | 24.50 | 23.45 | 21.36 | 21.05 |
| SSN-WRN-28-10 | **25.75** | **23.08** | **22.19** | **20.54** | **20.11** |
| % orig params | 32.8% | 46.5% | – | – | – |
| SWRN-28-10 (Savarese & Maire, 2019) | 20.48 | 20.04 | – | – | – |
| % orig params | 5.5% | – | – | – | – |
| SSN-WRN-76-12-9M | 19.16 | – | – | – | – |
| % orig params | 1.9% | 3.8% | 7.7% | – | – |
| DenseNet-BC (Huang et al., 2017) | 24.85 | 22.27 | 21.79 | – | – |
| SSN-DenseNet-BC-190-40 | **22.97** | **21.03** | **19.65** | – | – |
| % orig params | 9.7% | 17.9% | 50.0% | 75.1% | – |
| EfficientNet-ND (Ye et al., 2020) | 44.17 | 42.00 | 36.80 | 35.88 | – |
| SSN-EfficientNet-ND | **35.72** | **35.75** | **35.57** | **34.99** | – |

requirements by effectively sharing parameters. This enables us to train larger, better-performing networks than is possible with traditional neural networks on comparable computational resources.

## F IMAGE CLASSIFICATION NUMBERS

We provide raw numbers for the results in Figure 3 in Table 15 (CIFAR-100) and Table 16 (ImageNet).

## G PERFORMANCE IMPLICATIONS OF NPAS AND SSNS

Our SSNs can offer several performance benefits by reducing parameter counts; notably, they can reduce memory requirements storing a model and can reduce communication costs for distributed training. We emphasize that LB-NPAS does *not* reduce FLOPs, as the same layer operations are implemented using fewer parameters. Should fewer FLOPs also be desired, SSNs can be combined

Table 16: Performance of SSNs vs. traditional networks of a comparable size on image classification on ImageNet (from Figure 3 of the main paper). See Section 4.1 for discussion.

| Method | Performance E@5 (E@1) | | | |
|---|---|---|---|---|
| % orig params | 5.0% | 9.4% | 15.2% | 22.6% |
| WRN | 11.60 (31.81) | 10.02 (29.22) | 9.08 (27.58) | 8.44 (26.30) |
| SSN-WRN-50-2 | **11.07 (30.58)** | **9.02 (27.46)** | **8.10 (25.64)** | **7.42 (24.28)** |
| % orig params | 4.2% | 14.8% | 23.9% | – |
| EfficientNet | 10.84 (33.68) | 7.98 (28.47) | 7.53 (27.61) | – |
| SSN-EfficientNet-B0 | **10.21 (32.42)** | **6.82 (26.52)** | **6.44 (25.18)** | – |

with other techniques, such as pruning. Additionally, we note that our implementation has not been extensively optimized, and further performance improvements could likely be achieved with additional engineering.

### G.1 COMMUNICATION COSTS FOR DISTRIBUTED TRAINING

Communication for distributed data-parallel training is typically bandwidth-bound, and thus employs bandwidth-optimal allreduces, which are linear in message length (Chan et al., 2007). Thus, we expect communication time to be reduced by a factor proportional to the parameter savings achieved by NPAS, all else being equal. However, frameworks will typically execute allreduces layer-wise as soon as gradient buffers are ready to promote communication/computation overlap in backpropagation; reducing communication that is already fully overlapped is of little benefit. Performance benefits are thus sensitive to the model, implementation details, and the system being used for training.

For CNNs, we indeed observe minor performance improvements, as the number of parameters is typically small. When using 64 V100 GPUs for training WRN-50-2 on ImageNet, we see a $1.04\times$ performance improvement in runtime per epoch when using SSNs with 10.5M parameters (15% of the original model). This is limited because most communication is overlapped. We also observe small performance improvements in some cases because we launch fewer allreduces, resulting in less demand for SMs and memory bandwidth on the GPU. These performance results are in line with prior work on communication compression for CNNs (*e.g.*, Renggli et al., 2019).

For large transformers, however, we observe more significant performance improvements. The SSN-ALBERT-Large is about $1.4\times$ faster using 128 GPUs than the corresponding BERT-Large model. This is in line with the original ALBERT work (Lan et al., 2020), which reported that training ALBERT-Large was $1.7\times$ faster than BERT-Large when using 128 TPUs. Note that due to the differences in the systems for these results, they are not directly comparable.

We would also reiterate that for some applications where communication is more costly, say, for federated learning applications (*e.g.* McMahan et al. (2017); Konečný et al. (2016)), our approach would be even more beneficial due to the decreased message length.

### G.2 MEMORY SAVINGS

LB-NPAS and SSNs reduce the number of parameters, which consequentially reduces the size of the gradients and optimizer state (*e.g.*, momentum) by the same amount. It does not reduce the storage requirements for activations, but note there is much work on recomputation to address this (*e.g.*, Chen et al., 2016; Jain et al., 2020). Thus, the memory savings from SSNs is independent of batch size. For SSN-ALBERT-Large, we use 18M parameters (5% of BERT-Large, which contains 334M parameters). Assuming FP32 is used to store data, we save about 5 GB of memory in this case (about 1/3 of the memory used)

