# OpenReview forum: "Neural Parameter Allocation Search"
_ICLR.cc/2022/Conference — ICLR 2022 Poster_

### Official Review · Reviewer_cCHH · 2021-10-28

**Correctness:** 3
**Technical Novelty And Significance:** 3
**Empirical Novelty And Significance:** 3
**Recommendation:** 6
**Confidence:** 3

**Details Of Ethics Concerns:**




**Main Review:**

Figure 1. shows comparison between distillation and pruning which seems to be a different class of methods. It is mentioned to be complementary to NPAS. It would be better to show competitive analysis on prior work or similar approaches.

Slimmable networks is also a work presents a network that change parameters sizes automatically

NPAS adds training cost per epoch, but the parameter sharing enables faster convergence. Demonstrating that the effective train time to reach same accuracy would improve the paper.

The explanation flow in section 3 seems non-intuitive, because the method uses section 3.2 first and then it uses section 3.1.
Adding pseudo-code or algorithm would also improve clarity of the method.

page 8, typo: covolutional



**Summary Of The Paper:**

The paper presents a method to automatically select parameters to share between layers.
It proposes to use a shape shifter network to either increase or decrease the number of parameters in the model.
The parameters are mapped into parameter groups through a preliminary training step and k-mean cluster the layers.
Layers in the same group share parameters. It will generate weights by downsampling or upsampling depending on the layer needs.
The method is tested in Low Budget and High Budget regimes and on different tasks.
It also shows that the method can be used together with distillation and pruning.


**Summary Of The Review:**

The paper demonstrates SSN and NPAS and presents experimental results on various benchmarks.
The text could be improved for clarity and organization.

---

> ### Author Response · Authors · 2021-11-19
> **Response to reviewer cCHH**
>
> We thank the reviewer for their comments, we appreciate their time and will use their suggestions to improve our paper.
>
> **Figure 1. shows comparison between distillation and pruning which seems to be a different class of methods. It is mentioned to be complementary to NPAS. It would be better to show competitive analysis on prior work or similar approaches.**
>
> We note that our approach is an automated and generalized version of parameter sharing and we do include a comparison to prior work in parameter sharing in Figure 1. Pruning and distillation are also included since they share some similar goals to both the low budget and high budget NPAS settings as pruning and distillation, respectively. If there is a specific related task the reviewer thinks is not represented in Figure 1, we would be happy to respond to that suggestion.
>
> **Slimmable networks is also a work presents a network that change parameters sizes automatically**
>
> We note that slimmable networks work by training a model so that it can be executed with different widths on each forward pass.  These widths are predefined in slimmable networks as are which weights will be used to represent which layers, whereas our approach automatically learns where and how to share parameters between layers in a network.  In addition, slimmable networks do not reduce the number of parameters required to represent a specific network architecture, as our LB-SSNs can.   As a result, our approach can be used to save computational resources required for distributed training settings, whereas slimmable networks can not.  In fact, slimmable networks require significantly more computational resources per epoch, since each supported architecture requires a different forward pass through the model (as illustrated in Algorithm 1 of their paper).  In addition, our approach could be used in combination with a slimmable network, since they define the size of the layers and the target widths in a forward pass, but our approach could be used to generate the weights that are actually used to implement the slimmable network.  Thus, slimmable networks and NPAS/SSNs address different, but complementary tasks.

---

> > ### Author Response · Authors · 2021-11-19
> > **Additional response to reviewer cCHH**
> >
> > **NPAS adds training cost per epoch, but the parameter sharing enables faster convergence. Demonstrating that the effective train time to reach same accuracy would improve the paper.**
> >
> > While we have not conducted an expensive study on the convergence rates due to parameter sharing, and we would expect our approach would converge faster as that has been established for other parameter sharing techniques in prior work (e.g., ALBERT). We note that our approach does decrease training time in distributed learning settings.  As discussed in Section 2 of our paper, the overhead in a forward pass is negligible, and despite this we do see improved training speeds (discussed in more detail below).  The pretraining step does add some training time, but it is short and only needs to be executed once per architecture.  The learned mappings can then be stored and reused for any subsequent runs (e.g., when tuning hyperparameters).
> >
> > For CNNs, we observe a minor improvement to training speeds. When using 64 V100 GPUs for training WRN-50-2 on ImageNet, we observe about 1.04x performance improvements in runtime per epoch when using SSNs with 10.5M parameters (15% of original). This is limited because most communication is overlapped. When profiling, we also observe small performance improvements in some cases because we launch fewer allreduces, resulting in less demand for SMs and memory bandwidth on the GPU.
> >
> > For the BERT-Large architecture, the improvement is more significant, where SSNs are about 1.4x faster using 128 GPUs than the corresponding BERT-Large model.
> >
> > Finally, we note that we have not attempted to extensively optimize the performance of SSNs, and their runtime could likely be improved. In addition, we reiterate that for some applications where communication is more costly, say, for federated learning applications, our approach would be even more beneficial due to the decreased message length.  We have updated our paper to include the discussion on distributed training benefits in Appendix G.
> >
> > **The explanation flow in section 3 seems non-intuitive, because the method uses section 3.2 first and then it uses section 3.1. Adding pseudo-code or algorithm would also improve clarity of the method.**
> >
> > This organization was selected because the approach of Section 3.2 adapts methods that are described in Section 3.1.  The approach in Section 3.2 can simply be viewed as an offline pretraining step using a modified version of the approach described in Section 3.1.  I.e.,
> >
> > Step 1: Perform the pretraining step discussed Section 3.2 (which modifies the methods described in Section 3.1)
> >
> > Step 2: Use K-means to cluster the layer representations to create the parameter group mappings (described in the last paragraph of Section 3.2)
> >
> > Step 3: Use the parameter mapping from Step 2 to train our SSN (using the methods as they are described in Section 3.1)
> >
> > **page 8, typo: covolutional**
> >
> > Thank you for pointing this out! We have corrected the typo in our updated paper.

---

> > > ### Comment · Reviewer_cCHH · 2021-11-23
> > > **Response**
> > >
> > > I thank the authors for the response and clarification. They also addressed the concerns presented

---

### Official Review · Reviewer_p84E · 2021-11-02

**Correctness:** 3
**Technical Novelty And Significance:** 3
**Empirical Novelty And Significance:** 4
**Recommendation:** 8
**Confidence:** 4

**Main Review:**

#### **Strong points:**
1. This work is well-motivated. Parameter sharing is one of the most promising directions to reduce training and inference costs, but not addressed in AutoML community. So, the aim of the work is timely.
2. The proposed method is effective and efficient. Also, the authors validate the proposed method across not only multiple datasets but also multiple tasks. Moreover, the authors show that other techniques to reduce the inference cost are orthogonally applicable.

#### **Weak points:**
1. The proposed method contains many hyperparameters to be tuned, such as the window size, embedding dimension, and # partition groups. How sensitive is the performance of the proposed method to them? Also, do all tasks and datasets share the same set of hyperparameters?
2. The proposed method (LB-SSN) seems to reduce only the number of parameters. However, in the practitioner’s point of view, reducing inference time may be more critical. Are other pruning methods also applicable to the network configuration obtained by LB-SSN? It would be great to add the result to Table 4.

#### **Detailed comments:**
It is not easy to understand the proposed method with Figure 2 at a glance. Simplifying Figure 2 or replacing it with an abstractive algorithm description may be better.

**Summary Of The Paper:**

This work investigates how to automate allocating parameters within a fixed budget to each layer and generating its weights with assigned parameters, which is a generalized version of parameter sharing. The proposed method is validated with leading performances and efficiencies on multiple datasets across multiple tasks.

**Summary Of The Review:**

This work is well-motivated, and the experimental results support claims well. However, major concerns lie in the practices; the proposed method additionally introduces hyperparameters and cannot help practitioners reduce inference time.

---

> ### Author Response · Authors · 2021-11-19
> **Response to reviewer P84E**
>
> We thank the reviewer for their comments, we appreciate their time and will use their suggestions to improve our paper.
>
> **The proposed method contains many hyperparameters to be tuned, such as the window size, embedding dimension, and # partition groups. How sensitive is the performance of the proposed method to them? Also, do all tasks and datasets share the same set of hyperparameters?**
>
> We provide a sensitivity analysis for many of the hyperparameters in the Appendix.  Many of the experiments in our paper were the result of transferring default parameters or using a simple heuristic (e.g., ImageNet and ALBERT experiments).  In addition, most of these parameters do not make a significant difference in the performance in our experiments.  Specifically, the window size and embedding dimension are kept constant in all our experiments, demonstrating their ability to generalize.  When we tried different settings on CIFAR, they made negligible differences and were not tuned on any other datasets.  In fact, most settings can be set as the defaults we set on CIFAR, and the experiments on ImageNet and ALBERT both are examples of this process.  The only exceptions are the numbers of parameters and the number of parameter groups.  However, the number of parameters can be set based on the needs of a user, and the number of parameter groups have limited viable settings and can often be set via a simple heuristic that minimizes the amount of parameter upsampling required.
>
> To illustrate, note that the number of parameter groups can be, at most, equal to the number layers in a network.  As the number of desired parameters is lowered, then the number of parameter groups should be lowered accordingly. A WRN-28-10 network requires 36.5M weights for all its layers, and the largest layer accounts for a little under 4M of those weights.  Thus, if you were to train a model with fewer than 4M parameters, at least one layer would require upsampling even if you used just a single parameter group, so having much more than 2 would likely be very detrimental to performance since many layers would require weight upsampling.  As you raise the number of parameters available you could increase the number of parameter groups, but not so much that you would require significant upsampling.  This is illustrated in the first line of Table 12 in the appendix, where a higher number of parameter groups, which require significant parameter upsampling due to the relatively small number of parameters available, perform much worse than only a few groups.
>
> We note that when performing the pretraining stage to learn a mapping of layers to parameter groups, the only step that uses the number of parameter groups is when we cluster the layer representations (described in the last paragraph of Section 3.2).  This is key, because it means that from a single pretraining step we can produce all potential mappings by simply modifying the number of clusters we create.  Thus, you can use the heuristic of minimizing the amount of parameter upsampling required to filter out unsuitable mappings from a single pretraining step.  As shown in the second line of Table 12 of the appendix, when many parameters are available relative to the weights required by a network, there is only a small difference in performance between suitable candidate mappings.
>
> **The proposed method (LB-SSN) seems to reduce only the number of parameters. However, in the practitioner’s point of view, reducing inference time may be more critical. Are other pruning methods also applicable to the network configuration obtained by LB-SSN? It would be great to add the result to Table 4.**
>
> We note we already demonstrate that we can reduce inference time while improving performance using HB-SSN as the second column of Table 4.  We reiterate that in this setting we have no additional overhead from using a high budget at inference time, although we do have a small increase in the resources required for training the initial high parameter budget model.  That said, if you were to prune LB-SSN to 70% of the original test FLOPs using  HRank we found that Error@1 would be 9.43 (before pruning it was 9.26).  Given the relatively minor impact to performance, this suggests that low-budget SSN models may not be as negatively affected by parameter pruning as traditional models, but this requires more investigation.

---

> > ### Comment · Reviewer_p84E · 2021-11-22
> > **Response to the authors’ rebuttal**
> >
> > I thank the authors for the response.
> >
> > The authors addressed my concerns well. Also, the provided result about pruning LB-SSN is quite interesting. I recommend to add the result with the reason in the revision after more investigation. I’m happy to increase my score.

---

### Official Review · Reviewer_9sBX · 2021-11-02

**Correctness:** 3
**Technical Novelty And Significance:** 3
**Empirical Novelty And Significance:** 3
**Recommendation:** 6
**Confidence:** 4

**Main Review:**

The topic introduced in this paper is interesting and meaningful, which provides a novel idea for saving parameters. The proposed parameter upsampling and downsampling methods are promising.

Here are some questions to be answered.
1. The parameter number is indeed saved. However, how about the real performance on the hardware platforms, e.g., bandwidth, memory consumption, training speed? It is essential to report these results for evaluating the real value of the method.
2. How much benefit does the NPAS method bring? It is necessary to perform an experiment similar to "random search" in NAS to validate the effectiveness. Noting that in Fig. 4 (b), differences between the manual and auto one are not so evident, where it seems only the downsampling layers require unique parameters as in the manual one.
3. How are the parameters allocated in depthwise layers, e.g., in EfficientNet? Is there any difference between depthwise and plain convolutions?
4. Is the proposed method specified to tasks? It is interesting to explore whether parameters are allocated differently in different tasks.

**Summary Of The Paper:**

This paper proposes to solve an interesting but meaningful task, i.e., learning to allocate parameters between layers; in other words, searching for the parameter sharing strategy in a neural network. Parameter sharing between serial layers is useful for saving parameter numbers, which is beneficial for issues including memory consumption, bandwidth, etc.

**Summary Of The Review:**

The topic is interesting but the concerns listed above need to be cleared. The main concerns lie in the real-application value and real benefit of the methods.

---

> ### Author Response · Authors · 2021-11-19
> **Response to reviewer 9sBX**
>
> We thank the reviewer for their comments, we appreciate their time and will use their suggestions to improve our paper.
>
> **The parameter number is indeed saved. However, how about the real performance on the hardware platforms, e.g., bandwidth, memory consumption, training speed? It is essential to report these results for evaluating the real value of the method.**
>
> For CNNs, we observe a minor improvement to training speeds. Using 64 V100 GPUs for training WRN-50-2 on ImageNet, we observe about 1.04x performance improvements in runtime per epoch when using SSNs w/ 10.5M parameters (15% of original). This is limited because most communication is overlapped. When profiling, we also observe small performance improvements in some cases because we launch fewer allreduces, resulting in less demand for SMs and memory bandwidth on the GPU.
>
> For the BERT-Large architecture, the improvement is more significant, where SSNs are about 1.4x faster using 128 GPUs than the corresponding BERT-Large model.
>
> Finally, we note that we have not attempted to extensively optimize the performance of SSNs, and their runtime could likely be improved. In addition, we reiterate that for some applications where communication is more costly, say, for federated learning applications, our approach would be even more beneficial due to the decreased message length.
>
> Memory usage: SSNs reduce the storage requirements for parameters, which consequently reduces the storage requirements for gradients and optimizer state (e.g., momentum) by the same amount. It does not reduce the storage requirements for activations, but we note there is much work on recomputation to address this. For SSN-ALBERT-Large, we use 18M parameters (5% of BERT-Large, which contains 334M parameters). Assuming FP32 is used to store data, we save about 5 GB of memory in this case.
>
> We have updated our paper to include this discussion in Appendix G.
>
> **How much benefit does the NPAS method bring? It is necessary to perform an experiment similar to "random search" in NAS to validate the effectiveness.**
>
> We thank the reviewer for their suggestion.  We note that we already provided a comparison to random assignments in Appendix B.3 Table 9, where we demonstrate that searching for a good parameter mapping always performed better than random.  We have updated Table 7 to include random results from Table 9 corresponding to the relevant architectures in Table 7, but additional results on other architectures/tasks can still be found in Table 9.
>
> **Noting that in Fig. 4 (b), differences between the manual and auto one are not so evident, where it seems only the downsampling layers require unique parameters as in the manual one.**
>
> We note most downsampling layers in Fig 4(b) are shared with other layers using auto.  In auto later layers have their own parameters, but for manual many late layers share parameters. Also in Fig. 3(b) the manual approach SWRN-50-2 (prior work using manual parameter sharing techniques)  the lowest number of parameters achievable by their approach is around 40M parameters due, in part, to using manual mappings, whereas our SSNs get better performance using just over 15M parameters, 58% fewer than SWRN.  In addition, some architectures like DenseNets in Fig. 3(a) are not supported by the manual approach since they have no identical layers, i.e., SSNs generalize to more architectures than prior work.  Thus, this does indeed illustrate a significant benefit of our approach.
>
> **How are the parameters allocated in depthwise layers, e.g., in EfficientNet? Is there any difference between depthwise and plain convolutions?**
>
> No, there is no difference in our approach between depthwise and plain convolutions.   This highlights one of the benefits of our SSNs-  weight generation (Section 3.1) is agnostic in its operation to the type and size of layers, enabling us to support any layer by simply applying the same process.
>
> **Is the proposed method specified to tasks? It is interesting to explore whether parameters are allocated differently in different tasks.**
>
> We note that we are unclear as to whether the reviewer was asking if our approach can only be applied to certain tasks, or whether the same architecture may allocate parameters differently depending on the task.  In response to the first possibility, we note that our approach is not task specific and we provide experiments on nine network architectures and four different tasks, demonstrating that our approach generalizes.  For the second possibility, we trained a WRN-28-10 as a siamese network on the task of fashion compatibility using the Polyvore Outfits dataset from:
>
> Mariya I. Vasileva et al. Learning Type-Aware Embeddings for Fashion Compatibility. ECCV, 2018
>
> We found the parameter mapping assignments to be similar to the one in Figure 4(b).  Critically, the two observations we discussed at the end of Section 4.3 still held true.

---

> > ### Comment · Reviewer_9sBX · 2021-11-25
> > **Thanks for Your Response and Some Remained Concerns**
> >
> > Thank you so much for your detailed response, which has resolved most of my questions. However, I still have some concerns as follows.
> >
> > 1. The training speed acceleration is marginal for CNN networks but more evident for BERT-Large. What is the reason? For `we save about 5 GB of memory in this case`, could you please provide a more detailed description of this measurement, *e.g.* batch size and the original total memory cost? As for this state, I cannot figure out how significant "5 GB" is. From my perspective, it is essential to demonstrate how much benefit the proposed method can bring in real implementation considering elements listed in the review.
> > 2. For ImageNet classification in Tab. 9, "Emb" shows worse performance than "Random" and the advantage of "WAvg" is also marginal, which makes me doubtful about the effectiveness of the proposed method. By the way, what is the evaluation metric for ImageNet classification?
> > 3. `There is no difference in our approach between depthwise and plain convolutions.` Can the proposed method be applied to architectures with different types of convolution blocks in each layer, *e.g.* NAS-searched ones, as there may exist scenarios that parameters from depthwise convolutions are shared with plain ones?

---

> > > ### Author Response · Authors · 2021-11-26
> > > **Additional clarification**
> > >
> > > We thank the reviewer for their additional comments. We will incorporate them into the final version of the paper.
> > >
> > > **The training speed acceleration is marginal for CNN networks but more evident for BERT-Large. What is the reason?**
> > >
> > > This is primarily because CNNs have significantly fewer parameters than transformers, while being quite compute-intensive. For example, Wide ResNet-50-2 contains about 69M parameters while BERT-Large contains about 334M parameters. During backpropagation, frameworks will attempt to overlap communication and computation. In the case of CNNs, the communication volume is such that it can mostly be overlapped by computation, so performance improvements are limited.
> > >
> > > **For we save about 5 GB of memory in this case, could you please provide a more detailed description of this measurement, e.g. batch size and the original total memory cost? As for this state, I cannot figure out how significant "5 GB" is. From my perspective, it is essential to demonstrate how much benefit the proposed method can bring in real implementation considering elements listed in the review.**
> > >
> > > When training BERT, we use either a batch size of 96 for a sequence length of 128 or a batch size of 8 for a sequence length of 512 (depending on the phase of pretraining-- BERT initially uses the former then switches to the latter). These are the largest batch sizes that fit into memory on the V100 GPUs (with 16 GB of memory) we used. In both cases, the original memory cost is about 15 - 15.5 GB, so a reduction of 5 GB would reduce this by 1/3rd.
> > >
> > > However, note that the memory savings from NPAS/SSNs is actually independent of the batch size, because we only change the number of parameters, not the operations the layers compute. Thus, the benefit will grow for even larger models, while the overhead of activations can be handled by, e.g., recomputation.
> > >
> > > **For ImageNet classification in Tab. 9, "Emb" shows worse performance than "Random" and the advantage of "WAvg" is also marginal, which makes me doubtful about the effectiveness of the proposed method. By the way, what is the evaluation metric for ImageNet classification?**
> > >
> > > While the exact gains vary depending on the task and architecture, random assignments get the best performance in none of the 9 comparisons we make. In some comparisons, like ADAPT-T2I, random is 1% worse on COCO.  This clearly demonstrates a trend where learning the assignments is better than random assignments.  Although the gains on ImageNet are less than some other datasets, this is partly because it matters less when many parameters are available, as demonstrated that manual assignments get similar performance as WAvg.  However, as shown in Table 12, when fewer parameters are available there is a larger difference between the number of parameter groups, so is a case where good assignments may also be important as a poor set of assignments can result in many layers requiring significant upsampling. Finally, we note that our approach is only an initial baseline for our novel task and we expect that it can be improved upon. However, the benefits over prior work in parameter sharing is clear, outperforming SWRN on ImageNet using 60% fewer parameters (Figure 3b), which our approach actually uses fewer parameters than SWRN is even capable of, and getting a 1% gain over ALBERT (Table 2). We also note that parameter upsampling is another novel contribution of our work.
> > >
> > > As mentioned in Section 4.1, we use Error@5 to evaluate ImageNet classification.
> > >
> > > **There is no difference in our approach between depthwise and plain convolutions. Can the proposed method be applied to architectures with different types of convolution blocks in each layer, e.g. NAS-searched ones, as there may exist scenarios that parameters from depthwise convolutions are shared with plain ones?**
> > >
> > > We anticipate no issues in this setting. Indeed, note that, for example on the ADAPT-T2I network, we share parameters between very different kinds of layers (e.g., fully-connected and recurrent, as well as layers with different modalities), and most of our image classification experiments can share parameters between convolutional and fully connected layers.

---

> > > > ### Comment · Reviewer_9sBX · 2021-11-30
> > > > **Thanks for Your Response**
> > > >
> > > > I would appreciate the authors' detailed response, which has cleared most of my concerns. Here is still a minor issue: Error@1 is more commonly used for ImageNet evaluation, which is suggested to be used or supplemented. Besides, some contents in the appendix or rebuttal are very important and valuable for demonstrating the method's effectiveness. It is highly recommended to move these into the main text in the final version.

---

> > > > > ### Author Response · Authors · 2021-11-30
> > > > > **Further response**
> > > > >
> > > > > Thanks again for your comments. We will incorporate these updates into the final paper, including error@1 for ImageNet.

---

### Official Review · Reviewer_jt4B · 2021-11-04

**Correctness:** 3
**Technical Novelty And Significance:** 3
**Empirical Novelty And Significance:** 2
**Recommendation:** 5
**Confidence:** 3

**Main Review:**

Strengths

I commend the authors on extensive evaluation across multiple datasets, models, and tasks are convincing of the broad applicability of this approach. Not an expert in the literature, but during my investigation, the PNAS approach seems like a reasonably novel mechanism for parameter allocation and automatically learning which weights to share. Methodologically interesting approach, with implications for studies on generalizability.

Weaknesses

The practical impact of this approach is overclaimed in this paper. For example, while the SSN reach lower error compared to the EfficientNet family as the same # of trainable parameters, the EfficientNet family (and ALBERT, WRN, etc.) are designed to have lower inference time, which SSN does not achieve.

Figure 1 marks both the author's method (NPAS) and previous work in cross-layer sharing (presumably Savarasee & Maire, 2019) as "make training more efficient", but the effects are dramatically different. NPAS's efficiency is in lower memory footprint, whereas prior work reduces the FLOPs and therefore the training time (Table 4, Savarase & Maire, 2019).

While shared parameters does reduces the communication during data parallel training, the authors provide no data or reasoning for the magnitude of that effect on training time. Given that CNNs already have relatively low parameter count, I suspect that the effect on the model sync time during distributed training is minimal.

A meta-comment: parameter counts may be the wrong metric to compare model capacity in these studies, as the methods you compare against (e.g. the EfficientNet_B0 family) have reduce parameters by removing from network, whereas NPAS reduced by sharing weights. For example, would path norm be more appropriate from a generalization studies perspective, as hinted in your conclusion? This is admittedly out of scope for this paper, as the paper's main claims are on efficiency.

My score could be improved by:
* Figure 1: breaking out "makes training more efficient" into "Reduces FLOPs" and "reduces memory footprint" rows, and marking the methods appropriately. Citing the paper for prior work directly in the column or the caption.
* In the discussion of results, acknowledging that these other methods reduce FLOps and training time, whereas the NPAS approach does not.
* Providing data on the actual effect of this reduced number of trainable parameter counts on (1) model training times in a distributed training setting, and (2) memory usage on the GPU. Are these savings practically realizable?

**Summary Of The Paper:**

Parameter sharing can reduce memory footprint of neural networks and memory bandwidth requirements, but existing methods require manually tuning the sharing strategy. This paper uses a small phase of training to cluster the learned layer representations by groups. This allows networks to be scaled from small to large parameter counts (to be clear, number of trainable parameters) without changing the model architecture. Importantly, this procedure does not change the number of FLOPs in the model.

Experiments across a wide set of tasks and networks compare this approach with either (1) SWRN from Savarasese & Maire, 2019), or (2) existing hand-tuned parameter scaling from families of networks such as EfficientNet, DenseNet, or ALBERT.



**Summary Of The Review:**

The proposed method is methodologically an interesting contribution, however the practical impacts on training efficiency and model efficiency are minimal and overstated, which limits my recommendation.

---

> ### Author Response · Authors · 2021-11-19
> **Response to jt4B**
>
> We thank the reviewer for their comments, we appreciate their time and will use their suggestions to improve our paper.
>
>
> **The practical impact of this approach is overclaimed in this paper. For example, while the SSN reach lower error compared to the EfficientNet family as the same # of trainable parameters, the EfficientNet family (and ALBERT, WRN, etc.) are designed to have lower inference time, which SSN does not achieve.**
>
> We would begin by noting that we do get the improved inference time available to EfficientNet and WRN, since our SSNs can be used to implement those networks with low overhead (as stated in Section 2, for batch size 64 the overhead is between 0.008-0.03% of total FLOPs for a forward pass, and is lower for larger batches).  ALBERT is primarily a parameter sharing method, i.e., it reuses the same layer rather than using a new one for some operations.  While it does factorize embedding layers, this does not have a significant FLOP impact. As such, it does not reduce inference time, since the same operations are still performed, but only reduces the number of parameters in a network due to reusing layers.
>
> **Figure 1 marks both the author's method (NPAS) and previous work in cross-layer sharing (presumably Savarasee & Maire, 2019) as "make training more efficient", but the effects are dramatically different. NPAS's efficiency is in lower memory footprint, whereas prior work reduces the FLOPs and therefore the training time (Table 4, Savarase & Maire, 2019).**
>
> We note that Savarese & Marie is in fact a prior work in cross layer parameter sharing (as is ALBERT, which we compare to in Table 2).  One hallmark of parameter sharing methods, including these two, is that they do not reduce inference time, since the motivating idea is to reduce the computational resources (typically number of parameters, as is the case for Savarese & Marie, ALBERT, and our approach) by reusing a layer’s weights rather than creating new layers with their own weights.  Note that Savarese & Maire also have an overhead associated with their approach that is on par with our SSNs, i.e., they actually increase FLOPs, but the increase is negligible.
>
> We note that the experiment the reviewer is referring to in Table 4 of Savarese & Maire was comparing their models created with few parameters to networks with similar parameters created through neural architecture search (NAS).  In other words, they were comparing training using their parameter sharing approach to the time it would take using NAS to find one of a similar size.  It is well established that using architecture search takes longer than training a single network with a predefined architecture.  We reiterate that their training time is on par with training a standard neural network of the respective architecture when using a single GPU.  Below we provide the results of our approach as if they would be included as a row with the same settings as in Table 4 of Savarese & Maire:
>
> |       C-10         | Params (M) |   Training Time (GPU Days)  | Test Error  |
> |-----------------------:---------------------------|:----------------:|:----------------------------------:|:---------:|
> | SWRN-28-4 (Savarese & Maire)        |        2.7        |        0.06        |      4.06      |
> |  SSN-WRN-28-4 (ours)                      |        2.7        |       0.07         |     3.83       |
> | SWRN-28-6 (Savarese & Maire)        |        6.1        |       0.12         |     3.89       |
> | SSN-WRN-28-6 (ours)                       |         6.1        |       0.14        |      3.59     |
>
> In these experiments we used a single NVIDIA TITAN X GPU.  Note that Savarese & Maire report lower test error in Table 4 of their paper, but we find that we cannot reproduce those results using their publicly available code.  We further note that there has been an open issue on their Github page concerning the reproducibility of their Table 4’s results since November 2019 that has not been resolved.
>
> **While shared parameters does reduces the communication during data parallel training. Given that CNNs already have relatively low parameter count, I suspect that the effect on the model sync time during distributed training is minimal.**
>
> We have added a discussion of performance to the paper in Appendix G. To summarize, the impact of parameter sharing on communication is very model-dependent. While CNNs indeed show only a small performance improvement, the improvement is more significant with large models like SSN-ALBERT compared to BERT (discussed in detail later, but in line with prior results reported for ALBERT in Lan et al., 2020).
>
> Finally, we note that we have not attempted to extensively optimize the performance of SSNs, and their runtime could likely be improved. In addition, we reiterate that for some applications where communication is more costly, say, for federated learning applications, our approach would be even more beneficial due to the decreased message length.

---

> > ### Author Response · Authors · 2021-11-19
> > **Additional Response to jt4B**
> >
> >
> > **Figure 1: breaking out "makes training more efficient" into "Reduces FLOPs" and "reduces memory footprint" rows, and marking the methods appropriately. Citing the paper for prior work directly in the column or the caption.**
> >
> > We again reiterate that the prior work in parameter sharing such as SWRN and ALBERT does not reduce FLOPs, as we discussed earlier, so the suggested breakdown would not provide additional information.  We also note that this figure compares the tasks themselves and not specific methods. This is important since there are some methods that are more restrictive. For example,  some parameter pruning methods generalize to any architecture, while others may only support certain layers, e.g., there have been layer-specific methods introduced for convolutional or recurrent layers. As a result we selected the former in our comparison, as it is possible to have a generalized approach from prior methods, rather than the latter concerning specific methods.
> >
> > **In the discussion of results, acknowledging that these other methods reduce FLOps and training time, whereas the NPAS approach does not.**
> >
> > This is not the case, as we discussed earlier, i.e., none of the methods the reviewer suggested would reduce FLOPS or have a significant impact on training time compared to using a SSN.  In our paper we do acknowledge that parameter pruning methods, however, would reduce inference FLOPs and parameters, but they also significantly increase training time and the resources required since they require a fully parameterized network to prune.  In contrast, SSNs can train a network with minimal additional training time and with reduced memory/communication requirements, meaning our approach can make distributed training more efficient unlike pruning methods.  In addition, we can also reduce FLOPs and parameters as well by combining our approach and parameter pruning methods as we illustrate in the second column of Table 4. Finally, we note that in the HB-SSN setting, our approach also does create a network that needs fewer FLOPs compared to other models of a similar parameter size.
> >
> > **Providing data on the actual effect of this reduced number of trainable parameter counts on (1) model training times in a distributed training setting, and (2) memory usage on the GPU. Are these savings practically realizable?**
> >
> > Training times: Note that above we found that in a distributed training setup using 128 V-100 GPUs we achieved a 1.4 improvement when training a BERT-Large model and a 1.04x improvement when training WRN-50-2 on ImageNet.  We also again note for some applications communication costs are more expensive, such as federated learning, so our approach would be even more beneficial in these settings.
> >
> > Memory usage: SSNs reduce the storage requirements for parameters, which consequently reduces the storage requirements for gradients and optimizer state (e.g., momentum) by the same amount. It does not reduce the storage requirements for activations, but we note there is much work on recomputation to address this. For SSN-ALBERT-Large, we use 18M parameters (5% of BERT-Large, which contains 334M parameters). Assuming FP32 is used to store data, we save about 5 GB of memory in this case.
> >
> > We have updated our paper to include this discussion in Appendix G.

---

### Decision · Program_Chairs · 2022-01-20

**Decision:**

Accept (Poster)

**Comment:**

The reviewers were mostly concerned about the practical impact/implications of the proposed methods. There was a long discussion across multiple threads of the benefits of the approach proposed in CNNs vs larger language models, dissecting the benefits in terms of training time (as opposed to memory or FLOPs, which may have a non-linear impact on running time). Overall, the authors did a good job of putting their contribution into context and addressing the reviewer concerns.